# SYMMETRY AND SYSTEMATICITY

## ABSTRACT

We argue that symmetry is an important consideration in addressing the problem of systematicity and investigate two forms of symmetry relevant to symbolic processes. We implement this approach in terms of convolution and show that it can be used to achieve effective generalisation in three toy problems: rule learning, composition and grammar learning.

## 1 INTRODUCTION

Convolution (LeCun & Bengio, 1998) has been an incredibly effective element in making Deep Learning successful. Applying the same set of filters across all positions in an image captures an important characteristic of the processes that generate the objects depicted in them, namely the translational symmetry of the underlying laws of nature. Given the impact of these architectures, researchers are increasingly interested in finding approaches that can be used to exploit further symmetries (Cohen & Welling, 2016; Higgins et al., 2018), such as rotation or scale. Here, we will investigate symmetries relevant to symbolic processing.

We show that incorporating symmetries derived from symbolic processes into neural architectures allows them to generalise more robustly on tasks that require handling elements and structures that were not seen at training time. Specifically, we construct convolution-based models that outperform standard approaches on the rule learning task of Marcus et al. (1999), a simplified form of the SCAN task (Lake & Baroni, 2018) and a simple context free language learning task.

Symbolic architectures form the main alternative to conventional neural networks as models of intelligent behaviour, and have distinct characteristics and abilities. Specifically, they form representations in terms of structured combinations of atomic symbols. Their power comes not from the atomic symbols themselves, which are essentially arbitrary, but from the ability to construct and transform complex structures. This allows symbolic processing to happen without regard to the meaning of the symbols themselves, expressed in the formalist's motto as *If you take care of the syntax, the semantics will take care of itself* (Haugeland, 1985).

From this point of view, thought is a form of algebra (James, 1890; Boole, 1854) in which formal rules operate over symbolic expressions, without regard to the values of the variables they contain (Marcus, 2001). As a consequence, those values can be processed systematically across all the contexts they occur in. So, for example, we do not need to know who *Socrates* is or even what *mortal* means in order to draw a valid conclusion from *All men are mortal and Socrates is a man*.

However, connectionist approaches have been criticised as lacking this systematicity. Fodor & Pylyshyn (1988) claimed that neural networks lack the inherent ability to model the fact that *cognitive capacities always exhibit certain symmetries, so that the ability to entertain a given thought implies the ability to entertain thoughts with semantically related contents*. Thus, understanding these symmetries and designing neural architectures around them may enable us to build systems that demonstrate this systematicity.

However, the concept of systematicity has itself drawn scrutiny and criticism from a range of researchers interested in real human cognition and behaviour. Pullum & Scholz (2007) argue that the definition is too vague. Nonetheless, understanding the symmetries of symbolic processes is likely to be fruitful in itself, even where human cognition fails to fully embody that idealisation.

We investigate two kinds of symmetry, relating to permutations of symbols and to equivalence between memory slots. The relation between symbols and their referents is, in principle, arbitrary, and any permutation of this correspondence is therefore a symmetry of the system. More simply, the

names we give to things do not matter, and we should be able to get equivalent results whether we call it *rose* or *trandafir*, as long as we do so consistently. Following on from that, a given symbol should be treated consistently wherever we find it. This can be thought of as a form of symmetry over the various slots within the data structures, such as stacks and queues, where symbols can be stored.

We explore these questions using a number of small toy problems and compare the performance of architectures with and without the relevant symmetries. In each case, we use convolution as the means of implementing the symmetry, which, in practical terms, allows us to rely only on standard deep learning components. In addition, this approach opens up novel uses for convolutional architectures, and suggests connections between symbolic processes and spatial representations.

## 2 Rule Learning

The first symmetry to be considered is the one arising from the fact that the correspondence between referring atomic symbols and their referents is entirely arbitrary, which entails that permutations of these symbols are symmetries of the system. In fact, Corcoran & Tarski (1986) use this permutation invariance to define the difference between the logical and non-logical parts of a language.

This symmetry, in which any symbol is as good as any other, is anathema to the sort of problem that neural nets are typically applied to, in which the inputs are not arbitrary names but specific measurement values, e.g. images or medical records. In that case, effective learning requires discovering the appropriate differentiations, and would be sabotaged by randomly permuting the inputs.

Nonetheless, the work of Marcus et al. (1999) suggest that this symmetry may be relevant to human cognition, even for infants as young as 7 months. In these experiments, the infants were habituated to sequences of syllables which obeyed a simple rule, such as ABB (e.g. *la ti ti*) or ABA (e.g. *la ti la*). Subsequent testing on novel stimuli showed they were able generalise this rule to syllables not present in the training stimuli (e.g. *wo fe fe* or *wo fe wo*).

In other words the representation of the learned rule allowed it to be abstracted from the particular training stimuli and applied to any syllable. One interpretation is that the infants were treating the stimuli symbolically, in that one syllable was as good as any other, and that as a consequence their behaviour was symmetric under replacement of the syllables.

Marcus et al. (1999) were unable to obtain the same behaviour from a recurrent network architecture, because the statistical regularities it learned were linked to the specific syllables seen at training time and so generalisation to unseen syllables was not achieved. Here we show that this problem can be solved by imposing a symmetry on the architecture, that corresponds to weight sharing between syllables.

Practically, this is implemented as a one dimensional convolution of width one, followed by max-pooling across all syllables, and a softmax to produce output probabilities. The input consists of a $12 \times 3$ array of binary values representing the 12 syllables and 3 time steps, with convolution treating the syllables as positions and the time steps as channels. The output of the convolution has two channels, which after pooling become the logits for the binary outputs.

Note that this the opposite of how convolutions are most frequently used in application to language sequences. In that case, invariance to time translations is achieved by weight sharing across time steps, with the representation of each symbol being encoded in the channels. Here, in contrast, weight sharing happens between symbols and temporal information is encoded in the channels. This means our model is not invariant to translations in time, but is instead invariant to permutations of symbols.

Figure 1 shows the input sequence *wo fe wo* being processed by this architecture. The input is first encoded as activation in the first and third channels at the *wo* position, and activation in the second channel at the *fe* position. Convolution reduces these three channels down to two, and pooling projects this down to a pair of logits corresponding to the ABB and ABA categories.

The training and test inputs are taken from the Marcus et al. (1999) paper, and we train the model to distinguish ABB sequences from ABA sequences. We also train a multi-layer perceptron and a recurrent net on the same data, with the recurrence happening over the time dimension.

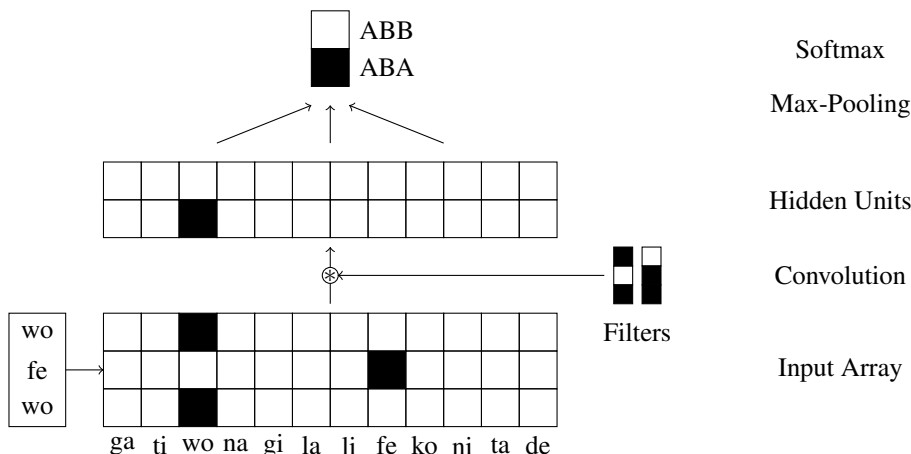

Figure 1: The architecture applied to the rule learning task of Marcus et al. (1999), consisting of convolution followed by max-pooling and softmax.

Table 1: Accuracies in identifying sequence structure

| Recurrent Net | Multi-layer Perceptron | Convolution |
|---|---|---|
| 50% | 50% | 100% |

The results on the test set in Table 1 show that neither the multi-layer perceptron nor the recurrent network learn a rule that generalises effectively to unseen syllables. However, the weight sharing in the convolutional net requires that the same function is applied to each syllable, giving perfect generalisation. The filter, being applied at every position, cannot discriminate between syllables, and instead can only respond to the information about temporal structure in the channels. So, for example, in the case of the sequence *wo fe wo*, the input channels at the *wo* position take the values 101, representing the fact that the same token occurs in the first and third temporal slots.

This is very similar to what Marcus et al. (1999) suggest is being learned by the infants: *algebra-like rules that represent relationships between placeholders (variables), such as 'the first item X is the same as the third item Y'*. Thus, by imposing a symmetry on the network, we learn functions that are sensitive to an abstract structure rather than the specific raw syllables in the input.

Thus, convolution provides a straightforward solution to this long-standing problem (Alhama & Zuidema, 2019) using the simple expedient of sharing weights between seen and unseen syllables. Moreover, we can make further use of the insight that symmetry allows us to abstract structure away from the particular content it contains.

## 3 COMPOSITION

Lake & Baroni (2018) investigated the systematicity of recurrent networks in terms of a task they call SCAN. This requires translating a sequence of instructions - such as *turn left and jump twice after walk* - into a sequence of actions - such as *LTURN WALK JUMP JUMP*. They found that while a sequence-to-sequence architecture could achieve near perfect scores when the test instances closely resembled the training data, performance broke down when generalisation outside the training distribution was required. In particular, the network struggled to translate sequences that were longer than those in the training data and those that contained novel combinations of words, e.g. *jump twice* when both *jump* and *twice* had been seen, but their combination had not.

A full solution for the SCAN task requires methods for handling a number of tricky problems in a systematic manner, e.g. parsing the instruction sequence, composing items, reversing the order of

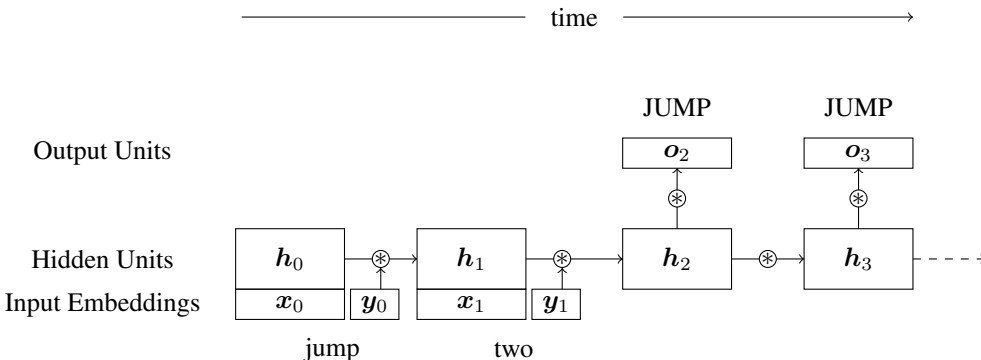

Figure 2: The encoder-decoder architecture applied to the composition task. For each input word, two embeddings are learned, one of which is concatenated with the current hidden state and the other forms the filter in a convolution applied to that input. In the decoder, actions are predicted by taking a convolution of the hidden state, and recurrence between time steps is also a convolution.

constituents linked by *after*. Instead, we focus on a single problem: that of learning to compose an instruction such as *jump* with a modifier such as *twice* in a way that generalises systematically.

Specifically, we construct a dataset in which the inputs are two word instructions drawn from a vocabulary of 10 commands (*jump, walk, run, ...*) and 10 modifiers (*one, two, three, ...*). Each instruction is to be translated into the corresponding action repeated the appropriate number of times, e.g. *jump four → JUMP JUMP JUMP JUMP*.

We randomly sample with replacement 1000 such translation pairs, choose three combinations and remove all instances of them from the training data and then exclusively test on these unseen pairings of command and modifier.

For a network to behave systematically, it must learn to associate a modifier, e.g. *four*, with a structure, e.g. repeat the same action four times, that applies to all actions indiscriminately. Following the lesson learned in the previous experiment, we might expect that the structural information might be best represented in the channels and filter of a convolution, and the actions as positions which are then treated symmetrically.

However, rather than hard-code this distinction into the encoder network innately, we allow a recurrent-convolutional architecture to discover this approach for itself. Each symbol is given two input representations, one of which is used as a filter in a width one convolution and the other of which feeds into the data that the convolution is applied to. Our intention is that the former should represent structural information (how many times to repeat) and the latter represent the content (which action) within that structure.

As shown in Figure 2, information flows through the model from these inputs to the output logits through a series of convolutions, which impose a permutation symmetry on the function learned in training. This invariance to permutations of the output symbols should permit the model to learn representations of structure which abstract away from the particular actions seen during training.

Within the recurrent-convolutional architecture, the hidden states, $h_t$, consist of an array of units comprising 5 channels by 11 positions, and one dimensional convolutions of width one are used as the basis for both recurrence between time steps and also to project the output logits from the hidden states. In the encoder, two embeddings $x$ and $y$ are learned for every word, one of which is concatenated with the hidden state, to become the 6th channel, and the other is used as the filter in the convolution that produces the next hidden state. In the decoder, two convolutional filters are learned, one of which projects the 5 hidden channels down to a single channel, to form the output logits, and the other predicts the new hidden state from the old at each time step. The targets are encoded as one-hot values in an 11-dimensional vector (10 actions + pad), and the loss is the cross entropy with the predictions. We also add an $L_1$ regularizer to the loss. Appendix B describes this architecture in more mathematical detail.

Table 2: Accuracies in translating instructions into actions.

| Recurrent Net | Multi-layer Perceptron | Convolution |
|---|---|---|
| 0% | 0% | 100% |

The results in Table 2 demonstrates that only the architecture exploiting the convolutional symmetry generalizes systematically. The net learns to associate a command, e.g. *jump*, with an action, e.g. *JUMP*, and a modifier, e.g. *two*, with an abstract structure, e.g. repeat the same thing twice. This is possible because the symmetry across symbols allows a structure to be represented in a way that makes no reference to the particular symbols instantiating that structure.

## 4   CONTEXT FREE LANGUAGE LEARNING

The structures considered in the previous section are extremely simple, having only short-range sequential dependencies. In contrast, the real grammars of natural languages produce long-range dependencies within hierarchical structures. Handling such structures, in which multiple dependencies are embedded within each other, will typically require some form of memory in order to keep track of the unresolved outer dependencies until the inner dependencies are completed. In this section we consider the role that symmetry plays in structuring this memory, again using convolution, which allows us to separate the contents of memory from the structure of how it is manipulated. In particular, we consider a reverse recall task that captures a key property of how such a memory has to operate.

For example, in the sentence *The racquet is actually very cheap* the subject noun, *racquet*, and main verb, *is*, display number agreement. In this case, both are singular, but they could also have been plural, i.e. *racquets* and *are*. For this simple sentence, noun and verb are adjacent so the span of the dependency is minimal. However, we can, in principle, insert as much material as we like, and this syntactic connection persists.

In particular, we can add a relative clause to the noun: *The racquet that the tennis player uses is actually very cheap*. Now another subject noun, *player*, and verb, *uses*, intervenes between the first pair, but the dependency, and in particular the number agreement, between *racquet* and *is* has to be maintained. This remains true even when we insert another relative clause: *The racquet that the tennis player we are all in awe of uses is actually very cheap*.

In this last case, the subject, *we*, and verb, *are*, are both plural, and effective processing of the whole sentence requires that this should not disrupt processing of the outer singular dependencies. In other words, a language user must be able to maintain a trace of multiple open dependencies until the relevant material is encountered. Notably, in the case of a centre embedded construction such as the sentence above, recall has to happen in a last-in-first-out manner as processing descends through the hierarchy and then rises back out again. That is, the subjects in the sentence above - *racquet, player, we* - are matched to verbs in reverse order - *are, uses, is*.

A common model for such structures are Context Free Grammars (CFG), which generate sentences in terms of production rules, such as those in Figure 3. These rules describe how the start symbol, $S$, is expanded into sequences of terminals symbols, such as *adoda* or *ccbabdodbabcc*. Each rule describes a substitution that can be applied to a single non-terminal symbol, i.e. $S$, $A$, $B$, $C$ or $D$ to yield a sequence of symbols. The context free aspect of such a grammar lies in the fact the substitutions are made without regard for the context around the original non-terminal symbol.

In the case of the grammar described in Figure 3, the substitutions applied to the non-terminals $A$, $B$, $C$ and $D$ yield a new string with the same terminal at the beginning and the end, and the final rule inserts an $o$. As a consequence, the resulting strings are palindromes with a single $o$ at their centre.

An equivalent model is a pushdown automata (PDA), in which hierarchical structure is handled by pushing symbols representing the outer structures onto a stack, until the interior structure is completed, and then popping symbols back off the stack to move outward in the hierarchy until no

$$
\begin{array}{ll}
S \to A & A \to aSa \\
S \to B & B \to bSb \\
S \to C & C \to cSc \\
S \to D & D \to dSd \\
& S \to o
\end{array}
\qquad
\begin{array}{ll}
S & \to A \\
& \to aSa \\
& \to aDa \\
& \to adSda \\
& \to adoda
\end{array}
$$

(a) Production rules for a simple CFG.      (b) Derivation of a palindrome.

Figure 3: A simple CFG, producing palindromic strings.

Table 3: Performance of LSTM Architectures on the palindromic language.

| LSTM Architecture | In-Domain | Long | $N_A > 2$ |
|---|---|---|---|
| Standard | 100% | 77% | 82% |
| Convolutional | 100% | 100% | 100% |

more symbols remain. Crucially, the stack has a last-in-first-out structure that essentially returns items in the reverse order in which they were pushed onto it.

In principle, such a system can handle sentences of unbounded length, containing arbitrarily long dependencies between constituents. In practice, however, language users struggle with nested structures more than two or three levels deep. Moreover, it is generally accepted that an ordinary CFG is not an accurate model of the grammatical structures that natural languages display. Instead, their grammars appear to be mildly context sensitive, as shown by evidence from Swiss German and Dutch (Shieber, 1985).

However, here we focus on CFGs and the ability of Recurrent Neural Nets to learn these structures. In particular, we investigate the ability of a Long Short Term Memory network to learn the simple palindromic language over the terminal symbols *a, b, c, d* and *o*, defined by the grammar in Figure 3.

We train an LSTM containing 100 hidden units, on 100,000 examples of strings of length 15, 17, 19, 21, 23 and 25, and then perform an in-domain test on novel strings of the same length, and an out-of-domain test on longer strings of lengths 29, 33, and 37. We also retrain the LSTM after removing all strings which contain more than 4 tokens of the symbol *a* from the training set, and then test only on examples from the test set containing more than 4 tokens of the symbol *a*.

The first row of Table 3 gives the results for this evaluation in terms of the proportion of symbols after the central *o* symbol that were predicted correctly. The in-domain results, in the first column, make it clear that the net has learned the structure of the grammar, and how to make accurate predictions, at least for sequences of lengths seen at training time. The second column, containing the results for longer sequences, shows that generalisation outside the range of the training set is not robust. The third column indicates that the model has difficulty generalising to sequences where the nesting of $A$ symbols is deeper than at training time, even though the actual length of sequences is unchanged.

These failures of generalisation can be seen as symptoms of the same underlying problem. In particular, success on each of these out-of-domain tasks simply requires extending the application of the same rules. However, the problem arises because the network lacks the required concept of sameness. The LSTM cells form an unstructured memory resource, without any notion of two cells containing the same information. Nor can there be a meaning to the idea of applying the same rule to that information. The parameters for each cell are learned independently, and so each cell carries out its own isolated task. During training these cells do learn to behave as a coherent whole, achieving impressive in-domain performance, but there is no way for the model to apply the same rule to the $n^{\text{th}}$ item as it applied to the previous $n-1$.

To address this shortcoming, we propose to organize the memory cells into an ordered one-dimensional stack structure, and to use convolutions to control the flow of information across timesteps, replacing the forget gate. The translational symmetry of this convolutional layer gives meaning to the idea of the same symbol being able to be stored in different cells and we use a filter

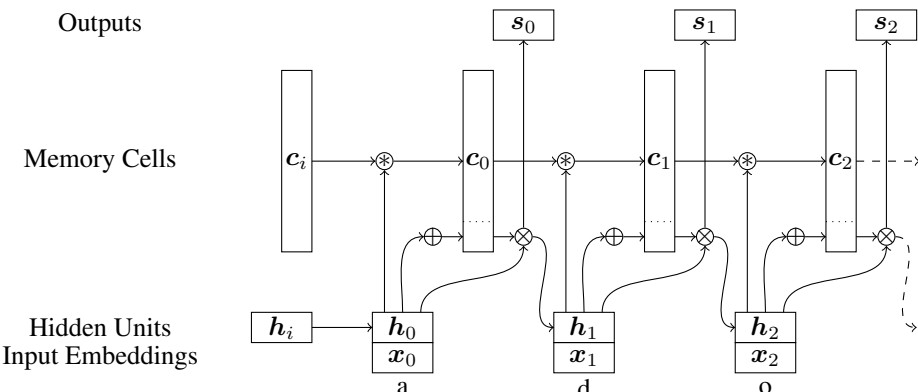

Figure 4: The modified LSTM architecture has a stack of memory cells which a convolutional gate controls the flow of information through, while a standard multiplicative output gate and additive input gate read and write to the cells in the bottom of the stack. As in a standard LSTM, the hidden units are concatenated with the input embeddings to form the vector that feeds into the units that drive the various gates.

of width three to allow the shifting of stored information between adjacent cells. Along with a restriction on input and outputs only being connected to the bottom of the stack, this turns the LSTM into a differentiable PDA. In particular, the filters $[1, 0, 0]$ and $[0, 0, 1]$ shift values back and forth when applied in a convolution and this allows the network to learn to shift memory contents up and down the stack in order to perform push and pop operations.

As shown in Figure 4, each token in the input is given an embedding which is then concatenated with the current hidden state. Together these values form the inputs to units that control the flow of information into, out of and between the memory cells, as in a standard LSTM. In this case, however, the memory cells are a set of one channel one dimensional convolutional layers and the forget gate has been replaced with a set of width three filters that shape the recurrent flow of information. These filters are the softmax outputs of units driven by the concatenated input embeddings and hidden units, allowing the network to use the input context to control the memory cell stack. Information is written to and read out from only the bottom entries in the stack, using standard input and output gates. This architecture is described in more mathematical detail in Appendix C.

Performance of this convolutional LSTM on the same evaluations is given in the second row of Table 3. There, all three columns show optimal performance on both the in-domain and out-of-domain tasks, demonstrating the utility of the convolutional layer in helping the model to generalise robustly.

## 5 RELATED WORK

Numerous authors have tackled the problem of replicating the rule learning behaviour studied by Marcus et al. (1999) in a connectionist system, and Alhama & Zuidema (2019) give an extensive review. Many of these approaches rely on a specific training regime to obtain the desired behaviour, rather than our approach of modifying the architecture to embed the appropriate capacities innately. However, our core intention was to demonstrate the relevance of symmetry, with convolution being a convenient and transparent means to that end. The same end could conceivably be achieved purely through learning.

The SCAN task of Lake & Baroni (2018) has also stimulated a number of responses which attempt to obtain the required systematicity. Lake (2019) uses a meta learning approach, which employed a training regime that explicitly permuted the correspondence between symbols and their meaning, i.e. between instructions and actions. This could be seen as learning an invariance to these permutations, rather than specifying it innately in the form of convolution. Russin et al. (2019), instead, take an approach based on learning separate semantic and syntactic representations for each word, which is comparable to our approach of learning two embeddings for each word.

The recurrent PDA we describe in Section 4 is very similar to a number of other architectures. Sun et al. (1998) proposed a neural network pushdown automata. Grefenstette et al. (2015) proposed architectures for a number of data structures: queues, dequeues and stacks. Joulin & Mikolov (2015) proposed a recurrent stack structure, which in practice, is almost equivalent to our proposal. However, none of these works discuss the role of symmetry or the connection to convolution.

Symmetries beyond spatial translation have been discussed by a number of authors. Cohen & Welling (2016) propose a generalisation of convolution for arbitrary discrete symmetries, such as reflections and rotations. The role of invariances in disentangled representations is discussed by Higgins et al. (2018), and Bloem-Reddy & Teh (2019) investigate the application of probabilistic symmetries to neural network architectures. Practical examples of symmetries supporting extrapolation and generalisation beyond the training set are discussed by Mitchell et al. (2018).

Permutation invariance is relevant to a number of representational strategies, such as bag-of-words approaches (e.g. White et al., 2015) or Deep Sets (Zaheer et al., 2017). However, the relevant symmetry in these cases is usually over permutations on the order of inputs, e.g. a symmetry between *wo fe fe* and *fe wo fe*. In our case, the permutation is over the identity of the symbols, i.e. a symmetry between *wo fe fe* and *la ti ti*.

## 6 DISCUSSION

One way to address the criticisms of distributed approaches raised by Fodor & Pylyshyn (1988) has been to focus on methods for binding and combining multiple representations (Smolensky, 1990; Hinton, 1990; Plate, 1991; Pollack, 1990; Hummel & Holyoak, 1997) in order to handle constituent structure more effectively. Here, we instead examined the role of symmetry in the systematicity of how those representations are processed, using a few simple proof-of-concept problems.

We showed that imposing a symmetry on the architecture was effective in obtaining the desired form of generalisation when learning simple rules, composing representations and learning grammars. In particular, we discussed two forms of symmetry relevant to the processing of symbols, corresponding respectively to the fact that all atomic symbols are essentially equivalent and the fact that any given symbol can be represented in multiple places, yet retain the same meaning. The first of these gives rise to a symmetry under permutations of these symbols, which allows generalisation to occur from one symbol to another. The second gives rise to a symmetry across memory locations, which allows generalisation from simple structures to more complex ones.

On all the problems, we implemented the symmetries using convolution. From a practical point of view, this allowed us to build networks using only long-accepted components from the standard neural toolkit. From a theoretical point of view, however, this implementation decision draws a connection between the cognition of space and the cognition of symbols.

The translational invariance of space is probably the most significant and familiar example of symmetry we encounter in our natural environment. As such it forms a sensible foundation on which to build an understanding of other symmetries. In fact, Corcoran & Tarski (1986) use invariances under various spatial transformations within geometry as a starting point for their definition of *logical notion* in terms of invariance under all permutations. Moreover, from an evolutionary perspective, it is also plausible that there are common origins behind the mechanisms that support the exploitation of a variety of different symmetries, including potentially spatial and symbolic. In addition, recent research supports the idea that cerebral structures historically associated with the representation of spatial structure, such as the hippocampus and entorhinal cortex, also play a role in representing more general relational structures (Behrens et al., 2018; Duff & Brown-Schmidt, 2012).

Thus, our use of convolution is not merely a detail of implementation, but also an illustration of how spatial symmetries might relate to more abstract domains. In particular, the recursive push down automata, discussed in Section 4, utilises push and pop operations that relate fairly transparently to spatial translations. Of course, a variety of other symmetries, beyond translations, are likely to be important in human cognition, and an important challenge for future research will be to understand how symmetries are discovered and learned empirically, rather than being innately specified.

A common theme in our exploration of symmetry, was the ability it conferred to separate content from structure. Imposing a symmetry across symbols or memory locations, allowed us to abstract

away from the particular content represented to represent the structure containing it. So, for example the grammar rule learned by our network on the syllable sequences of Marcus et al. (1999) was able to generalise from seen to unseen syllables because it represented the abstract structure of ABB and ABA sequences, without reference to the particular syllables involved. We explored how this ability could also be exploited on composition and grammar learning tasks, but it is likely that there are many other situations where such a mechanism would be useful.

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

## APPENDICES

## A   SYMMETRY

Informally, a symmetry of a system is a mapping of the system onto itself which preserves the fundamental properties of the system. In the case of translation symmetry, we have input images, $x$, and output labels, $y$, and we want to learn a function, $f(x)$, which predicts these labels, and is invariant to spatial translations, $T$. That is, we want $f$ to obey $f(Tx) = f(x)$.

Typically, we achieve this by composing two types of function: equivariant convolutions, $c$, and invariant poolings, $p$. Equivariant here means that the output from a translated input is itself the translation of the original output: $c(Tx) = Tc(x)$. While invariant means the output is unchanged by input translations: $p(Tx) = p(x)$.

When the width of the convolution is reduced to one, the function, $c()$, becomes equivariant to all permutations, $S$, not just translations: $c(Sx) = Sc(x)$. Permutation equivariance arises, for example, in formal logic, where the rules of deduction depend not on the particular names within an expression, but on its logical structure. So, *Socrates is mortal* follows from *Socrates is a man* and *All men are mortal* not because of the meaning of *Socrates* or *mortal*, but because the syllogism has the right form. Thus, if $x$ represents the premises and $y$ represents the conclusions, then the process of deduction $d(x) = y$ should be equivariant under any substitution, $S$, of names. That is, we should be able to reach an equivalent conclusion even if we rename *Socrates* as *Bob*, and so $d(Sx) = Sd(x)$.

Symmetries also arise in computational processes. For example, if $x$ is the state of some machine containing an addressable memory and running a program that references various addresses then we want the behaviour of the machine to be equivalent if we utilise a different set of memory addresses. That is, the state transition function, $f()$, should be equivariant under permutations, $S$, of states, which apply equivalent permutations of the memory content and the addresses referenced in the program: $f(Sx) = Sf(x)$.

## B   COMPOSITION

Our approach to the composition task is based on the encoder-decoder architecture shown in Figure 2. This is made more precise in the equations below.

In the encoder, each token in the input is given an N-dimensional embedding, $\boldsymbol{x}$ and an M+1 $\times$ M dimensional embedding, $\boldsymbol{y}$. The former is concatenated to the current N $\times$ M dimensional hidden state to become the M+1th channel of a vector, $\boldsymbol{g}$.

$$\boldsymbol{g}_t = \boldsymbol{h}_t \oplus \boldsymbol{x}_t \tag{1}$$

A convolution of width one is applied to this vector to generate the next hidden state, using the second embedding as filter.

$$\boldsymbol{h}_{t+1} = \boldsymbol{y}_t \circledast \boldsymbol{g}_t \tag{2}$$

The decoder takes the final hidden state from the encoder and projects it down to a single channel, using a width one convolution. These are then the logits of a softmax output, $\boldsymbol{o}$, predicting the next action.

$$\boldsymbol{o}_t = \text{softmax}\left(\boldsymbol{f}_o \circledast \boldsymbol{h}_t\right) \tag{3}$$

The next hidden state is also produced by a width one convolution, in this case maintaining the number of channels.

$$\boldsymbol{h}_{t+1} = \boldsymbol{f}_r \circledast \boldsymbol{h}_t \tag{4}$$

An $L_1$ regularisation is added to the cross entropy between the predictions, $\boldsymbol{o}$, and a one hot encoding, $\boldsymbol{a}$, of the correct actions.

$$Loss = H(\boldsymbol{a}, \boldsymbol{o}) + \lambda_x |\boldsymbol{x}|_1 + \lambda_y |\boldsymbol{y}|_1 \tag{5}$$

For the composition task, we used N=11 and M=5.

## C   CONTEXT FREE LANGUAGE LEARNING

Figure 4 gives a visual overview of the architecture we apply to learning the simple palindromic language. This is essentially a modified LSTM in which the forget gate has been replaced with convolutional filters. We define this explicitly below.

Each token in the input is given an M-dimensional embedding, $\boldsymbol{x}$, which is concatenated with the current N-dimensional hidden state, $\boldsymbol{h}$, to give a vector, $\boldsymbol{g}$, representing the current context.

$$\boldsymbol{g}_t = \boldsymbol{h}_t \oplus \boldsymbol{x}_t \tag{6}$$

This controls $N$ single channel width three filters, $\boldsymbol{f}$, each of which is the output of a softmax.

$$\boldsymbol{f}_{n,t} = \text{softmax}\left(\boldsymbol{W}_{f,n}\boldsymbol{g}_t + \boldsymbol{b}_{f,n}\right) \tag{7}$$

As in a standard LSTM, values written to the cells are the outputs of tanh units gated by a sigmoid.

$$\boldsymbol{i}_t = \sigma\left(\boldsymbol{W}_i \boldsymbol{g}_t + \boldsymbol{b}_i\right) \circ \tanh\left(\boldsymbol{W}_c \boldsymbol{g}_t + \boldsymbol{b}_c\right) \tag{8}$$

And the output gate is also a sigmoid function.

$$\boldsymbol{o}_t = \sigma\left(\boldsymbol{W}_o \boldsymbol{g}_t + \boldsymbol{b}_o\right) \tag{9}$$

Recurrence between the cells in each of the $N$ memory stacks is based on one-dimensional convolution and controlled by the filters $\boldsymbol{f}$.

$$\boldsymbol{c}_{n,t} = \boldsymbol{f}_{n,t} \circledast \boldsymbol{c}_{n,t-1} \tag{10}$$

The $0^{\text{th}}$ values in each stack are updated using the values of $\boldsymbol{i}$.

$$c_{n,t,0} = c_{n,t,0} + i_{t,n} \tag{11}$$

And the new hidden state are also read out from the $0^{\text{th}}$ values, gated by tanh units.

$$h_{t+1,n} = o_{t,n} \cdot \tanh\left(c_{n,t,0}\right) \tag{12}$$

The final outputs, to predict a one hot vector representing the next symbol, apply a softmax to these new hidden states.

$$s_t = \text{softmax}\left(W_s h_{t+1} + b_s\right) \tag{13}$$

The loss is the cross entropy over the second half of the sequence, and for this task M and N are both 10, with the memory stack having a depth of 20.

