# OpenReview forum: "Symmetry and Systematicity"
_ICLR.cc/2020/Conference — Reject_

### Official Review · AnonReviewer3 · 2019-10-22
**Official Blind Review #3**

**Rating:** 1

**Review:**


======================================== Update after rebuttal =============================================

I have now read the author rebuttal, but my concerns about the paper remain. The training details are not described in anywhere near sufficient detail (optimizer?, batch size?, learning rate?, initialization?, etc). The baseline architectures “recurrent net” or the “multi-layer perceptron” are not described at all, despite my explicit request to that effect. I had also requested to see the source code for the experiments as this would perhaps have illuminated a lot of the details left out in the paper, but the authors have not provided it. I understand that the authors are not required to provide their code, but this should have been a relatively straightforward request in this case given the simplicity of the experiments and as I mentioned in my initial review, it would have been very useful in evaluating the paper.

In their rebuttal, the authors also claimed that the results in Fig. 1 and Table 1 are training results (that even though the architecture is "innate", the weights are learned), but I'm concerned about this claim. I happen to be doing some experiments along these lines at the moment, and it is not trivial at all to get such crisp results as those shown in Fig. 1 & Table 1 in these kinds of experiments (even when the architecture is correctly specified). Again, it would have been very helpful if the authors had either provided their source code or had described their experimental setup in sufficient detail to allow the reproduction of these results.

Given these concerns, I have decided to keep my score as it is.

========================================================================================================

This paper addresses an important problem: systematic generalization in neural networks. However, the paper is very confusing and I have some serious concerns about the models and the results presented in sections 3 and 4. Here are the main issues:

1) In section 3, there are only 10x10=100 possible combinations in this composition task. Yet, the paper says “we randomly sample 1000 such translation pairs, choose three combinations and remove all instances of them from the training data and then exclusively test on unseen pairings of command and modifier.” How can you sample 1000 pairs out of a possible 100 combinations? Also being able to generalize to 3 held-out combinations out of 100 is not very impressive. On the contrary, it is almost trivial.

2) No details are given about the “recurrent net” or the “multi-layer perceptron” baselines in section 3. What are these models? The fact that they have exactly zero accuracy is a bit suspicious, especially given the almost trivial nature of the task in section 3. Previous works reported perfect or near perfect accuracy with similar baselines in similar tasks (see e.g., Lake & Baroni, 2018).

3) I'm afraid the proposed model in section 3 also doesn’t make sense to me. It is explicitly acknowledged (Appendix B) that y is an M+1-dimensional vector, g is an Nx(M+1) matrix. Then by all accounts, the convolution of these should be an N-dimensional vector. Yet, somehow, h_t+1 in Equation 2 manages to be an NxM matrix. How? Please clarify this. If possible, making the source code available would be very helpful.

4) What is the semantics of x and y in section 3? What exactly are they supposed to be doing? This is not explained in the paper beyond a vague description.

5) Similar problems arise in section 4. The task is not explicitly described in the text. We only learn from Appendix C that it is actually to predict the next symbol. The task description mentions “strings of length 15, 17, 19, 21, 23 and 24,...” (p. 5). But, the grammar in Fig. 3 can only generate odd length strings, it cannot generate a string of length 24. Is this a typo?

6) Again in section 4, I have no idea how the proposed model is actually supposed to work. The motivation for the model and its description are not clear at all.

7) The model in section 2 is hand-coded. It is not shown that it can actually learn this solution from data. What happens if the sequences are longer or if the rules are different, for example? Then you have to hand-code a completely different architecture.

8) Which brings me to another important issue I have with this paper (and with similar papers): this whole set-up is very misguided in my mind. I think the real problem is not to come up with an architecture that would generalize systematically in a very specific (and usually toy) problem. It is to come up with an architecture that would learn to generalize systematically in a much broader set of problems. The learning aspect in sections 3-4 is a step in the right direction, but there’s no evidence that the models proposed there can learn to generalize in any task other than the very specific tasks they were designed for (if they can actually do that).

**Experience Assessment:**

I have read many papers in this area.

**Review Assessment: Checking Correctness Of Derivations And Theory:**

N/A

**Review Assessment: Checking Correctness Of Experiments:**

I carefully checked the experiments.

**Review Assessment: Thoroughness In Paper Reading:**

I read the paper at least twice and used my best judgement in assessing the paper.

---

> ### Author Response · Authors · 2019-11-08
> **Typos and Clarifications from Review #3**
>
>  Thank you for identifying these issues.
>
> > How can you sample 1000 pairs out of a possible 100 combinations?
>
> We sample with replacement.
>
> > y is an M+1-dimensional vector ...  Please clarify this.
>
> This is a typo. It should be an (M+1)xM dimensional vector.
>
> > “strings of length 15, 17, 19, 21, 23 and 24,...” (p. 5). ...  Is this a typo?
>
> Yes, it should be 25.

---

> ### Author Response · Authors · 2019-11-08
> **Discussion of Models and Experiments**
>
> Thanks for raising these questions.
>
> > Also being able to generalize to 3 held-out combinations out of 100 is not very impressive. On the contrary, it is almost trivial.
>
> Training on 97% of the data excludes 'not having seen enough of the data' as an explanation for a failure to generalise. And this is precisely why we chose this regime, as this allow us to focus on the impact of symmetry.
>
> The results suggest that the problem is trivial for the convolutional architecture, but almost impossible for the MLP and RNN.
>
> It is trivial for the convolutional architecture, because the symmetry allows it to represent structure (e.g. repeat the same thing twice) independently of the content (e.g. JUMP). So when it encounters 'jump two' at test time it has no problem generalising. The MLP and RNN in contrast learn hidden representations that are typically conjunctive, ie do not represent structure and content separately. So it is very unlikely for them to generalise robustly.
>
> However, the difficulty or triviality of the task is not the point of the experiment, which is instead to investigate how symmetry can be used to support composition.
>
> > Previous works reported perfect or near perfect accuracy with similar baselines in similar tasks (see e.g., Lake & Baroni, 2018).
>
> Although our task is a simplification of SCAN, this does not mean our task is easier. SCAN is complex because it contains a diverse range of structures, some of which are easier than others.
>
> For example, 'jump left and look left twice' occurs in one of the SCAN test sets, while 'jump' on its own and 'walk left and look left twice' occur in the corresponding training set. Generalisation in this case is easier, because 'jump' and the context it occurs in can be translated independently.
>
>
> > What is the semantics of x and y in section 3?
>
> As suggested in paragraph 5 of Section 3, our intention is to represent an action (e.g. JUMP) in terms of a position and a structure (do it twice) in terms of the channels of a convolutional network. So, x represents the action to be performed and y is the structure.
>
> > I have no idea how the proposed model is actually supposed to work. The motivation for the model and its description are not clear at all.
>
> The basic idea is that CFGs allow nested structures, e.g. a noun phrase contained within another noun phrase. If the same rules apply to all noun phrases, however deeply embedded, then this is a kind of symmetry. In particular, for an LSTM to handle these structures we really need a symmetry over the memory cells, so that it can hold multiple constituents of the same type, as it moves through a nested structure.
>
> A convolution over the memory cells not only supplies the symmetry that makes the idea of the same symbol stored in multiple places meaningful, it also introduces the possibility of shifting symbols across the stack of memory cells. This is important because a context free language can also be defined in terms of a PDA. The stack of the PDA has push and pop operations that correspond to shifting everything one place further into the stack and writing a new symbol at the top, or reading a symbol from the stack and shifting everything one place back. For convolution, these shift operations can be defined in terms of width 3 filters.

---

> ### Author Response · Authors · 2019-11-08
> **Discussion of Methodology**
>
> Thanks for raising these concerns.
>
> > The model in section 2 is hand-coded. It is not shown that it can actually learn this solution from data.
>
> The architecture is designed for this particular task, but the weights are learned from the data. This is not an unusual setup. Nor are convolution and pooling anomalous architectural choices.
>
> But the point of the paper is not to sell a particular architecture, it is to investigate how symmetries relevant to symbolic processes can be introduced into neural architectures, and whether that leads to more systematic generalisation.
>
> > It is to come up with an architecture that would learn to generalize systematically in a much broader set of problems.
>
> I agree that this should be a core objective of Machine Learning. And I am happy to concede that I have not presented in this paper an architecture that faithfully reproduces the full robustness of human generalisation capabilities.
>
> Toy problems and simple architectures help us to progress towards this goal, because they allow us to investigate specific questions under controlled conditions with comprehensible models.
>
> All of the models described in this paper are shallow and built out of standard components, and while the reviews suggested this simplicity or lack of novelty was a problem they also managed to be confused by what exactly these apparently trivial models were doing. I am unconvinced a deeper and more sophisticated architecture applied to more complex and heterogeneous tasks would have been easier for readers to understand and so shed more light on the questions posed.
>
> > there’s no evidence that the models proposed there can learn to generalize in any task other than the very specific tasks they were designed for
>
> It is fairly common for ML papers to focus on a single dataset, i.e. to provide no evidence that the particular innovation they introduce is relevant to any other task.
>
> In contrast, we showed that symmetry was relevant to obtaining systematic generalisation in three different tasks. We also discussed how the symmetries imposed on the neural architectures related to the properties of symbolic systems. In other words, we provided a range of both theoretical and experimental evidence that imposing the right kind of symmetries can support systematic generalisation.

---

### Official Review · AnonReviewer2 · 2019-10-23
**Official Blind Review #2**

**Rating:** 1

**Review:**

The paper investigates the idea of using symmetry and invariance in symbolic reasoning. In particular, it considers models where modeling symbolic symmetries through parameter-sharing help with generalization. The three tasks considered are: 1) rule learning: performs sequence-classification. 2) composition: performs sequence-to-sequence with structured input using encoder-decoder architecture, and 3) context-free language learning using memory. In the first two cases, the convolution (of single width) is used to benefit from the symmetry prior, and in the third task, convolution is applied to stack memory structure. In all cases, the proposed architectures were shown to outperform MLP and RNN.

The paper addresses an important area in deep learning, and the paper is accessible and easy to read. However, there are major issues:

-- I found it challenging to identify a novel contribution. For example, the first task is simply using a single convolution layer followed by pooling for sequence prediction. However, using 1D convolution layers are somewhat wide-spread in NLP.

-- The paper is oblivious to a large body of related work in the area of relational learning and invariant/equivariant deep learning. Here are some examples: Permutation invariant model for sets [1,2], and the link between parameter-sharing and invariance [3,4] is theoretically studied in several works. Note that convolution with a filter of width one followed by pooling is exactly invariant to the symmetric group. There are related works that extend these ideas to graphs [5,6], and relational learning [7]. Invariance has also been explored as it relates to memory [8]. Another relevant direction to discussions of the paper is the idea of attention in various architectures, such as transformers.

-- There are vague or misleading claims. In particular, for some tasks, it is not clear why the proposed architecture addresses the targeted symmetry. For example, it is not clear why translation invariance in-memory models the structure in a context-free grammar.


[1] Zaheer, Manzil, et al. "Deep sets." Advances in neural information processing systems. 2017.
[2] Qi, Charles R., et al. "Pointnet: Deep learning on point sets for 3d classification and segmentation." Proceedings of the IEEE Conference on Computer Vision and Pattern Recognition. 2017.
[3] Shawe-Taylor, John. "Building symmetries into feedforward networks." 1989 First IEE International Conference on Artificial Neural Networks,(Conf. Publ. No. 313). IET, 1989.
[4] Ravanbakhsh, Siamak, Jeff Schneider, and Barnabas Poczos. "Equivariance through parameter-sharing." Proceedings of the 34th International Conference on Machine Learning-Volume 70. JMLR. org, 2017.
[5] Kondor, Risi, et al. "Covariant compositional networks for learning graphs." arXiv preprint arXiv:1801.02144 (2018).
[6] Maron, Haggai, et al. "Invariant and equivariant graph networks." arXiv preprint arXiv:1812.09902 (2018).
[8] Kazemi, Seyed Mehran, and David Poole. "RelNN: A deep neural model for relational learning." Thirty-Second AAAI Conference on Artificial Intelligence. 2018.
[9] Vinyals, Oriol, Samy Bengio, and Manjunath Kudlur. "Order matters: Sequence to sequence for sets." arXiv preprint arXiv:1511.06391 (2015).


**Experience Assessment:**

I have read many papers in this area.

**Review Assessment: Checking Correctness Of Derivations And Theory:**

N/A

**Review Assessment: Checking Correctness Of Experiments:**

I assessed the sensibility of the experiments.

**Review Assessment: Thoroughness In Paper Reading:**

I read the paper at least twice and used my best judgement in assessing the paper.

---

> ### Author Response · Authors · 2019-11-07
> **Response to Review #2**
>
> Thank you for your comments.
>
> > For example, the first task is simply using a single convolution layer followed by pooling for sequence prediction. However, using 1D convolution layers are somewhat wide-spread in NLP.
>
> This is a misunderstanding. Our application of convolution is significantly different to the standard application to sequences in NLP.
>
> In the standard use, weight sharing happens across time steps, and the function learned is equivariant to time-translations. In our case, weight sharing is between symbols, and the function is equivariant to permutation of symbols.
>
> So, in Figure 1, the standard way of applying convolution to sequences would take the horizontal syllable dimension as the channels and convolve in the vertical dimension of time steps. In our network, this is reversed, and we take time steps as the channels and convolve across symbols, as explained in paragraph 6 of section 2.
>
> The standard convolutional approach to sequences (weight sharing across time) will not solve Marcus's challenge, but our approach does.
>
> However, the point of the experiment is not to show off a new architectural innovation. It is to demonstrate that imposing a permutation invariance on symbols (as discussed by Tarski) allows us to model the rule learning behaviour studied by Marcus.
>
> > The paper is oblivious to a large body of related work
>
> Thanks for the references, some of which may usefully expand our related work section. However, many of these papers are only distantly or superficially related to the problems and architectures we discuss in the paper.
>
> > it is not clear why translation invariance in-memory models the structure in a context-free grammar
>
> The relation between CFGs and PDAs is well known, and imposing translation invariance on the memory cells turns an LSTM into a PDA. In particular, the push and pop operations of a PDA can be thought of as translation within the stack of memory cells. If we apply the width 3 filter 001 to a vector of values, this will shift all the values one place to the left, while 100 corresponds to a right shift. Along with only reading and writing to one end of the stack, this reproduces the behaviour of a PDA.
>
> More abstractly, a symmetry across memory locations allows us to treat all instances of a symbol equivariantly. This allows the architecture to exploit memory slots at test time that were not used in training. As a consequence, the network more readily extends the learned grammar to more complex examples (i.e. those requiring more memory), essentially because the symmetry gives meaning to the idea of applying the same rule to all memory slots.

---

> > ### Comment · AnonReviewer2 · 2019-11-13
> > **clarification regarding related works and contribution**
> >
> > Thanks for your response and explanations.
> >
> > Please let me clarify my objection to the dismissal of related work:
> >
> > The first experiment as pointed out is using convolution with a filter of width one, which is the permutation invariant model that is studied in the related works I have cited (i.e., the input is treated as a bag of words.) Changing the width of this convolution to multiple words will make the model invariant to the translation of symbols. Both of these types of parameter-sharing models in language are studied in previous works. The paper ignores these works. Applying invariant architectures to toy problems that are by design permutation invariant, is not by itself a significant contribution (the final experiment is an exception to this, where unfortunately the claims are precise.)
> >
> > I disagree with the statement that these recent advances in building invariant networks are "distantly or superficially related". This work is about the application of such invariant networks to language.
> >
> > Thanks for clarifications on the final experiment. I suggest making the statement about the role of symmetry in memory more precise using math. As it is, I find the reasoning hard to follow.

---

> > > ### Author Response · Authors · 2019-11-15
> > > **Relation between bag-of-words, Deep Sets and our approach.**
> > >
> > > Thanks for your response and suggestions.
> > >
> > > >  the input is treated as a bag of words
> > >
> > > This is a misunderstanding. A bag-of-words approach would not be able to learn to distinguish ABB from ABA on the training set, never mind generalise to the test set. Under a bag-of-words approach, wo fe fe looks the same as fe wo fe, so ABB and ABA are indistinguishable.
> > >
> > > The permutation we consider in our paper is not permutation within the input, but permutation of the symbols themselves, i.e. wo is replaced with la. As I explained above, we do not share weights across time steps (as in a bag-of-words approach) but instead between symbols.
> > >
> > > Thus we are talking about a different sort of permutation, having considerably different applications, from bag or set representations. Indeed, that form of permutation is not really relevant to the issue of systematicity. Thus, these works are only superficially related to this paper.

---

> > > > ### Comment · AnonReviewer2 · 2019-11-15
> > > > **model of a set**
> > > >
> > > > Thanks for the clarification. You are right, the first model is not simply a bag of words, rather a bag of word+locations.
> > > >
> > > > (The multi-hot coding used for the representation of the input is encoding the presence/absence of the word and its location in the sentence.)

---

### Official Review · AnonReviewer1 · 2019-10-24
**Official Blind Review #1**

**Rating:** 3

**Review:**

This paper focuses on modelling invariances or symmetry between various components for solving tasks via convolutions and weight sharing.
The proposed tasks are toyish in nature although they do give insights into importance of modeling symmetry for better generalization. The first task is a symbol substitution which considers a permutation in source symbols and maps them to either "ABA" or  "ABB" categories i.e binary classification. While this task does require generalizability, it is surprising that the mlp and recurrent net baselines are so much inferior (basically random) to the convolution baseline. While this shows efficacy of modeling symmetry, I'd be curious about performance graphs as the training data increases in size.
The second task is an artificially created task inspired from the SCAN dataset. The task is to translate a verb-number pair into number repetitions of the verb. The encoder decoder network uses convolution in the recurrences to capture the notion of generalizability. The input and output space is very small (10 verbs and 10 numbers) but shows superiority of convolution and weight sharing over other baselines. Curiously, the recurrent baseline seems to perform better than 0% accuracy (if still poorly) on the original SCAN task which is much harder than the proposed task in this paper. Maybe, the number of examples (1000) is too small recurrent networks but this makes me a little surprised. More details about the architecture and training procedure for baselines would be helpful to ensure that the comparison is fair across baselines.
The final task is CFG modeling where convolutions are used to model the forget gate of an LSTM which seems to endow the network with PDA like properties and the convolutions are more effective than baselines at modeling this.

Apart from the concerns related to the results mentioned above, my major concern is that the tasks considered are too simple and at least one complicated large-scale task would have strengthened the paper.
Also, for tasks 2 and 3, the motivation behind using convolutions is not as clean as in task 1. So more analysis and insights into model performance, the weights learned, ablation studies etc. would have helped in understanding how the convolutions are modeling the symmetry. This should be informative and tractable because of simplicity of the tasks involved.

Finally, as mentioned above, I still cannot intuitively understand why convolutions in the forget architecture would learn about symmetry related to structured repetition produced by a CFG. Hence, more analysis or a better motivation would have helped.

**Experience Assessment:**

I have published one or two papers in this area.

**Review Assessment: Checking Correctness Of Derivations And Theory:**

I carefully checked the derivations and theory.

**Review Assessment: Checking Correctness Of Experiments:**

I carefully checked the experiments.

**Review Assessment: Thoroughness In Paper Reading:**

I read the paper thoroughly.

---

> ### Author Response · Authors · 2019-11-13
> **Response to Review #1**
>
> Thanks for your comments.
>
> > While this shows efficacy of modeling symmetry, I'd be curious about performance graphs as the training data increases in size.
>
> The key point about Marcus's task is that the syllables encountered at test time are not present in the training data. So, given a one-hot representation, the units representing those unseen syllables are never active during training and in a standard MLP or RNN that means the weights connected to those units are never updated. No additional quantities of training data change this, and their performance remains at random.
>
> Convolution solves this problem by sharing weights between syllables, allowing an abstract pattern to be learned across all syllables. Test time performance on the unseen syllables is identical to the other syllables, because the symmetry ensures all the syllables are equivalent.
>
> There are other ways to solve this problem, for example by changing the representation. But this avoids the central problem of learning rules that can be applied to novel inputs. Our solution imposes a symmetry of symbolic systems onto the net and as a consequence learns abstract rules of the kind that Marcus was interested in.
>
>
> > Curiously, the recurrent baseline seems to perform better than 0% accuracy (if still poorly) on the original SCAN task which is much harder than the proposed task in this paper.
>
> Actually, many of the instances in the original SCAN test sets are easier than ours.
>
> For example, 'jump left and look left twice' occurs in one of the SCAN test sets, while 'jump' on its own and 'walk left and look left twice' occur in the corresponding training set. Generalisation in this case is easier, because 'jump' and the context it occurs in can be translated independently.
>
> > I still cannot intuitively understand why convolutions in the forget architecture would learn about symmetry related to structured repetition produced by a CFG.
>
> In a width 3 convolution, the filter 100 corresponds to shifting values to the right and 001 corresponds to shifting to the left. In the stack of memory cells, these shifting operations can be used to move the contents up and down the stack. Along with reading and writing only from the bottom of the stack this mimics a Push Down Automata, and the correspondence between PDAs and CFGs is well known.
>
> More concretely, to predict palindromes like aadcbobcdaa the net learns to push the first half of the string onto the stack until it reaches the central o and then switches direction to pop items off the stack until the end.
>
> > my major concern is that the tasks considered are too simple and at least one complicated large-scale task would have strengthened the paper.
>
> Would a complicated large-scale task have made things clearer?
>
> Ultimately, we intend to apply the ideas described here to a more naturalistic task, e.g. language modelling. However, real data brings a host of complications that can obscure the underlying hypotheses. Here we have introduced the idea that symmetry can be a means to gain systematicity, and demonstrated this on three toy problems, which allowed us to investigate specific questions in a controlled manner.
>
> We found that symmetry allows us to generalise to unseen symbols, to separate content from structure and to generalise to structures that are more complex than those seen at training. If we had rejected the use of controlled experiments then it is unlikely that we could have gained comparable clarity on these questions.

---

> > ### Comment · AnonReviewer1 · 2019-11-14
> > **Thanks for the response but my main concerns still linger**
> >
> > Thanks to the authors for their response and further explanation of results observed in tasks 1 and 2. Regarding cfgs, I would like to see a more formal mathematical connection behind the motivation to use cnns in the final draft.
> > My main concern is the lack of experiments on a large scale task. Right now, it looks like the paper focuses on just 3 systemic rules that are desirable to encode in the learning network and proposed different instantiations of convolutional neural networks to encode the three specifically identified phenomena. Is there a more general purpose network that would account for all the three phenomena and many of the other much more complex systematic rules? Also, does the proposed technique that learns to encode a specific form of systematicity finally help in  improving performance on a real world task. My suggestion is not to reject the controlled experiments but to add to the current set of experiments. If the proposed architectures are indeed efficient at encoding such useful systematic rules then we should expect to see improvements w.r.t. performance or sample complexity on real world task. Even results on full SCAN dataset would go about throwing some light on the effectiveness of reasoning about systemic rules.
> > Also, for the 3 different tasks, although the architectures are based on CNNs, they are very different from each other and need to be manually designed. How feasible is this to use CNNs for capturing more complex systematic rules in the real world datasets?
> > I'm not inclined to change my score based upon the current draft and the author response.

---

### Decision · Program_Chairs · 2019-12-19

**Decision:**

Reject

**Comment:**

Thanks for clarifying several issues raised by the reviewers, which helped us understand the paper.

After all, we decided not to accept this paper due to the weakness of its contribution. I hope the updated comments by the reviewers help you strengthen your paper for potential future submission.